# Elevated Temperature Effects on Protein Turnover Dynamics in *Arabidopsis thaliana* Seedlings Revealed by ^15^N-Stable Isotope Labeling and *ProteinTurnover* Algorithm

**DOI:** 10.3390/ijms25115882

**Published:** 2024-05-28

**Authors:** Kai-Ting Fan, Yuan Xu, Adrian D. Hegeman

**Affiliations:** 1Agricultural Biotechnology Research Center, Academia Sinica, Taipei 115, Taiwan; kaitingfan@sinica.edu.tw; 2MSU-DOE Plant Research Laboratory, Michigan State University, East Lansing, MI 48824, USA; 3Departments of Horticultural Science and Plant and Microbial Biology, University of Minnesota, Twin Cities, MN 55108, USA

**Keywords:** ^15^N-stable isotope labeling, crop resilience, *Arabidopsis thaliana*, heat stress, protein turnover, proteomics

## Abstract

Global warming poses a threat to plant survival, impacting growth and agricultural yield. Protein turnover, a critical regulatory mechanism balancing protein synthesis and degradation, is crucial for the cellular response to environmental changes. We investigated the effects of elevated temperature on proteome dynamics in *Arabidopsis thaliana* seedlings using ^15^N-stable isotope labeling and ultra-performance liquid chromatography-high resolution mass spectrometry, coupled with the *ProteinTurnover* algorithm. Analyzing different cellular fractions from plants grown under 22 °C and 30 °C growth conditions, we found significant changes in the turnover rates of 571 proteins, with a median 1.4-fold increase, indicating accelerated protein dynamics under thermal stress. Notably, soluble root fraction proteins exhibited smaller turnover changes, suggesting tissue-specific adaptations. Significant turnover alterations occurred with redox signaling, stress response, protein folding, secondary metabolism, and photorespiration, indicating complex responses enhancing plant thermal resilience. Conversely, proteins involved in carbohydrate metabolism and mitochondrial ATP synthesis showed minimal changes, highlighting their stability. This analysis highlights the intricate balance between proteome stability and adaptability, advancing our understanding of plant responses to heat stress and supporting the development of improved thermotolerant crops.

## 1. Introduction

Global warming threatens plant immunity, endangering global food supply and ecosystems [1,2], with predictions suggesting a decline in crop yields by 11–25% by the century’s end [3]. In light of this warming trend, the development of crop varieties engineered for enhanced thermotolerance is imperative to safeguard food production [4]. Heat stress manifests observable phenotypes at the whole plant level, including suppressed seed germination, inhibited shoot and root growth, fruit discoloration, leaf senescence, and diminished yield [5,6]. At the cellular level, heat stress induces physical changes such as increased membrane fluidity and protein denaturation, thereby affecting protein synthesis, enzyme activity, and metabolism [7]. Interestingly, moderate heat stress, around 28 °C, can trigger phenotypes suggesting improved evaporative cooling capacity despite elevated water loss and transpiration rates [8].

Photosynthesis, particularly Photosystem II (PSII), is significantly affected by heat stress [9,10], with moderate heat causing PSII photoinhibition [11], while higher temperatures can lead to dissociation or inhibition of the oxygen-evolving complex [12]. Although Rubisco, the enzyme responsible for carbon fixation, is intrinsically thermostable in higher plants, heat stress can inhibit Rubisco activase, thereby impacting carbon assimilation rates [13,14]. Notably, Rubisco activase acts as a major limiting factor in plant photosynthesis under heat stress, with the introduction of thermostable Rubisco activase variants resulting in enhanced carbon assimilation rates under moderately high temperatures [13].

Plants employ diverse molecular mechanisms to adapt to elevated ambient temperatures [15,16]. Elevated temperatures elevate the concentration of misfolded, unfolded, and aggregated proteins, triggering the transcriptional activation of heat stress-induced genes [17]. Among these genes are various families of heat shock proteins (HSPs), which act as molecular chaperones regulating protein folding and stability [18]. The unfolded protein response (UPR) in plants represents a vital signaling pathway in response to stress, initiating processes including protein translation attenuation, activation of the ER-associated degradation pathway, and induction of endoplasmic reticulum (ER) chaperones [19]. Heat stress not only affects protein stability but also disrupts specific enzyme functions, thereby perturbing metabolism [20]. Additionally, oxidative stress accompanies the heat stress response, resulting in the accumulation of reactive oxygen species (ROS). Coping with ROS accumulation and other oxidative stress injuries represents a significant challenge for organisms experiencing heat stress [21]. ROS production triggers an antioxidant response mediated through a MAPK signal pathway and the induction of downstream transcription factors. A key aspect of this response involves the removal of ROS molecules using ROS-scavenging enzymes such as ascorbate peroxidase (APX) and catalase (CAT) [17].

In the realm of genomics, researchers have identified thousands of genes that may be differentially regulated at the transcriptional level in response to heat stress in various plant species, including *Arabidopsis* [22,23], tomato [24,25], rice [26,27], barley [28,29], wheat [30,31], and maize [32,33]. However, the steady-state levels of transcripts do not fully reflect the levels of corresponding proteins, as translation serves as a crucial point of regulatory control in the plant heat stress response [34,35]. These studies underscore the inadequacy of solely relying on transcriptional analyses of the heat response in plants.

In this study, we conducted a proteome-wide analysis to monitor changes in protein turnover in *Arabidopsis thaliana* seedling tissues following exposure to elevated temperature (30 °C). This research introduces, to our knowledge, a novel approach aimed at evaluating, for the first time, the dynamic balance of protein synthesis and degradation in response to moderate heat stress in intact plant seedlings. We utilized the publicly accessible algorithm *ProteinTurnover*, enabling us to execute an automated pipeline to measure protein turnover rates through the ^15^N-metabolic stable isotope labeling approach on a proteomic scale [36]. As demonstrated in a prior study [37], this algorithm, when combined with stable isotope labeling, offers the capability to explore the comprehensive scope of metabolism, encompassing metabolic rates/fluxes and the static pool size of plants.

The goal of this study was to evaluate the impact of moderate heat treatment on protein turnover rates across various cellular fractions of Arabidopsis seedling shoots and roots at a proteomic scale. To achieve this aim, seedlings were transferred to media containing the stable isotope ^15^N and subjected to heat stress (30 °C), while seedlings under control conditions (22 °C) were continuously grown on ^14^N-medium. Root and shoot tissues were collected at different time points (0, 8, 24, 32, and 48 h) post-transfer and then subjected to differential centrifugation to isolate fractions enriched in organellar, soluble, or microsome-associated proteins for analysis by LC-MS/MS. Our study identified hundreds of proteins exhibiting significant changes in turnover rates in response to elevated temperature stress across root or shoot soluble, organellar, and microsomal fractions. This highlights adaptive proteomic mechanisms in response to heat stress, providing insights for enhancing crop resilience and productivity in the face of global climate change.

## 2. Results

### 2.1. Peptide Identification and Selection Criteria for Protein Turnover Rate Measurements

From the root tissue, 822 and 857 proteins were identified in the enriched soluble fraction from the control and 30 °C groups, respectively. In the enriched organelle fraction, 494 and 377 proteins were identified from the control and 30 °C groups, respectively. Additionally, 1222 and 1054 proteins were identified in the enriched microsomal fraction from the control and 30 °C groups, respectively. Thousands of identified peptides were required for the subsequent turnover analysis due to the lower sensitivity inlet used in this study. As indicated in Appendix A, each sample contained between 5000 and 14,000 peptides, but only 30–50% of them were present at a sufficient number of time points to compute turnover rates. In this dataset, peptides were most frequently excluded because they were not identified in the time 0 dataset.

Applying multiple quantitative quality criteria for the inclusion of each peptide can enhance the quality of the resulting turnover data and accelerate data processing. Peptides with significant standard errors typically represent those with poor spectral fitting, often due to co-eluting contaminants (Figure 1A). Peptides were included in further analysis if they met specific criteria: a visual score for spectral fitting (to the beta-binomial model) greater than 80 out of 100, a standard error in the turnover rate fitting of less than 10, and data points for at least three of the time points (including time 0). These criteria were chosen based on an empirical visual inspection of peptide turnover fitting plots generated by the algorithm. Additionally, the normal quantile-quantile (Q-Q) plot of peptide log_2_*k* was utilized to assess whether the log_2_*k* data were normally distributed (Figure 1B,D). Scatter plots of log_2_*k* and the standard error of log_2_*k* (such as shown in Figure 1A,C) aided in assessing dataset quality. Inspection of Figure 1C also suggests a potential negative linear correlation between log_2_*k* and the standard error of log_2_*k*, at least for this dataset. Nonetheless, only peptides selected using the aforementioned filtering criteria were used for further turnover rate analysis. Once a peptide passed this filter, it was assumed that the turnover rate calculated for each peptide contributed equally to the final protein turnover rate. Therefore, the log_2_*k* of all selected peptides was averaged to yield each individual protein turnover rate (log_2_*k*) for a given experimental condition (control vs. treatment).

### 2.2. Overview of the Effects of Heat Stress on Peptide and Protein Turnover Rates

#### 2.2.1. Trends in Peptide or Protein Turnover Rates

The distributions of peptide turnover rates (log_2_*k*) between the control and 30 °C groups are depicted for comparison purposes as histograms for soluble, organellar, and microsomal protein-enriched fractions of shoot and root tissues in Figure 2. The distributions of protein turnover rates (log_2_*k*) between the control and 30 °C groups are illustrated for comparison purposes as histograms for soluble, organellar, and microsomal protein-enriched fractions of shoot and root tissues in Figure 3. When comparing the mean values of peptide turnover rates or the median value of protein turnover rates between roots and shoots, generally across all fractions, the turnover rates of roots were faster than those of shoots. The average protein turnover rate (log_2_*k*) was −5.308, −5.594, and −5.377 in the soluble, organellar, and microsomal fractions, respectively, while in shoots, the average protein turnover rate was −6.0348, −6.1046, and −5.9765 in the soluble, organellar, and microsomal fractions, respectively. For the control group, the mean protein turnover rates (log_2_*k*) were close to −5.39 in roots and −6.03 in shoots, indicating that the mean protein half-lives were 29.13 h in roots and 45.2 h in shoots, suggesting that the root proteome might have a faster turnover rate than the shoot proteome in general. This may be related to the development of root tissue in the young seedling stage of plants, which requires more rapid changes in protein synthesis and degradation.

As the mean may be a more robust population estimator than the median for the bimodal distribution, the mean value was shown in each peptide rate distribution in Figure 2. In every fraction of root or shoot tissue, the average peptide log_2_*k* of the 30 °C group was less than that of the control, indicating that peptides tend to turn over faster in response to a higher temperature. The difference in the mean log_2_*k* between the control and 30 °C was about 0.17 in the root enriched soluble fraction, 0.18 in the root organelle enriched fraction, 0.25 in the root microsomal enriched fraction, 0.41 in the shoot soluble enriched fraction, 0.30 in the shoot organelle enriched fraction, and 0.33 in the shoot microsomal enriched fraction. Therefore, there was a 1.12~1.18-fold change in the turnover rate of root peptides and a 1.23~1.32-fold change in the turnover rate of shoot peptides at the elevated temperature. At the level of proteins, the fold change of average turnover rate due to 30 °C stress ranged from 1.16 in the root enriched soluble fraction, ~1.31 in the root organelle enriched fraction, 1.22 in the root microsomal enriched fraction, 1.26 in the shoot soluble enriched fraction, 1.23 in the shoot organelle enriched fraction, and 1.34 in the shoot microsomal enriched fraction. Both peptide and protein turnover rate distributions in the three protein fractions indicate that shoot and root proteomes have different scales of response to high temperatures. Comparing the change in protein turnover rate between roots and shoots in response to high temperatures using ANOVA and Tukey’s HSD test revealed a significant difference in log_2_*k* (*p* < 0.001).

The histograms of some data groups exhibit bell-shaped distributions with slightly asymmetrical patterns in both control and treatment groups. It is possible that the bimodality at the peptide level reflects variations in amino acid content, which could influence peptide turnover rate calculations. In general, the presence of bimodality is less apparent in the protein turnover histograms (Figure 3) compared to the peptide histograms (Figure 2). This observation is not surprising given the significant decrease in the number of observations from peptides to protein turnover. One potential method to test for bimodality is by employing Hartigan’s dip test [38]. In the dip test, the null hypothesis states that the distribution of the sample is unimodal, while the alternative hypothesis suggests that the distribution is not unimodal, indicating at least bimodality. The results from the dip test indicated a significant non-unimodal or at least bimodal distribution of peptide turnover rate (*k*) in the control group of the root microsomal fraction (*p*-value = 0.00376) and marginally non-unimodal in the root organellar fraction (*p*-value = 0.0847).

#### 2.2.2. Coefficient of Variation in Protein Turnover as a Function of the Number of Peptide Observations

Figure 4 shows the extent of variation in protein turnover in this experiment as a function of the number of peptide observations that were averaged to produce the rate for each protein. Since the protein turnover rates were obtained as the mean turnover rates of all selected peptides, the coefficient of variation (CV), also known as relative standard deviation, can be used to show variability in relation to the mean of the population. Here, the values of CV were calculated as the standard deviation divided by the absolute value of protein turnover rate log_2_*k*. Comparing Figure 4A,B, it appears that both the control and 30 °C datasets have similar levels of variability, suggesting consistency in the protein turnover rates between these two groups. At first, it appears as though the CV values for the protein turnover rates are larger for the rate values calculated from smaller numbers of peptides, but the median CV ranged from 0.02 to 0.05 and is independent of peptide number. The illusion of high CV for small numbers of peptides is due to the inverse correlation between the number of rates calculated and the number of peptides used for each calculation. As a result, there are significantly more real outliers for the very well-defined distribution of CV of protein turnover rates from two peptides. Most CV values are within the range of 0 to 0.10, while less than ~10 proteins have a CV greater than 0.10. When only two peptides were computable for one protein, there were only three or four cases where the CV was greater than 0.15. Given this analysis of CV, it is quite reasonable to include proteins with turnover rates calculated from as few as two computable peptides and to make protein turnover rate comparisons between samples with different numbers of computable peptides.

#### 2.2.3. Statistical Significance of Changes in Protein Turnover Rates upon Heat Treatment

Proteomic analysis of protein turnover requires a large number of individual UHPLC-HRMS/MS analyses to provide data across multiple time points, different tissues, different biochemical fractions, and test conditions. These analyses take a considerable amount of time and are expensive. For this reason, it is often impractical to use sampling of biological replicates as a means of testing statistical significance. Furthermore, these analyses often fail to identify many of the lower abundance proteins in replicate runs due to the element of chance in precursor ion detection. As a result, replicated peptide observations are only available for a portion of the identified proteins, and typically only those in the top several orders of magnitude in protein abundance. Given the time, cost, and repeatable coverage considerations, a reasonable alternative for determining the significance of changes in turnover rate (log_2_*k*) between treatments is to apply a linear mixed-effect model (LMM) [39]. An LMM allows one to estimate the likelihood of a difference in log_2_*k* values between treatments using a linear model consisting of a mixture of fixed and random effects. The fixed effects represent the errors associated with the conventional linear and non-linear regression portions of the turnover rate derivation, and the random effects represent unknown but random effects such as how peptides were selected from the population of peptides during the UHPLC-HRMS/MS analysis. The LMM approach is also compatible with taking the average of the peptide turnover rate values to determine the protein turnover rate. Appendix A lists the output of the LMM estimation.

A summary of the number of identified peptides and proteins in this study, with the applied threshold for selection, and their number with significant changes in turnover rate (log_2_*k*) due to the 30 °C treatment (*p* < 0.05) identified in the enriched soluble, organellar, and microsomal fractions of *Arabidopsis* seedling root or shoot tissues are listed in Appendix A. The identified proteins with significant changes in turnover rate (log_2_*k*) are listed in Appendix A, with at least one unique peptide in both control and 30 °C samples, which were discussed further (Figure 5, Figure 6, Figure 7 and Figure 8). An overview of the distributions of estimated differences in protein turnover rates between control and heat stress is shown in Figure 5 as histograms (Figure 5A) or box plots (Figure 5B). Overall, proteins enriched in the shoot soluble fraction had the largest change in turnover rate, with a median increase of ~0.492 log base 2 scales, or ~1.41-fold increase in protein turnover rate (*k*) upon heat stress. The box plots in Figure 5B demonstrate that all but the root or shoot soluble fraction had similar variations in the change in protein turnover rate upon heat stress. ANOVA and Tukey’s HSD tests revealed that there was a significant difference in the fold change of turnover rate between root and shoot soluble fractions (*p* < 0.001). There were also differences between shoot soluble and shoot organellar fractions (*p* < 0.01) and root soluble and root microsomal fractions (*p* < 0.05). It suggests that the proteins in the shoot tissue exhibit a greater change in rates of turnover in response to high temperatures than the proteins in the root tissue. Hence, the root proteome may not be as responsive as the shoot proteome to temperature change.

### 2.3. Links between Protein Functional Categories and Changes in Protein Turnover Rates upon Heat Treatment

#### 2.3.1. Protein Function and Turnover Rates of Proteins

In a comparison of the shoot and root soluble fractions, the proteins in shoots exhibited a much higher change in turnover rates than in the roots in response to elevated temperature (Figure 5). To determine if function might play a role in protein stability, root and shoot proteins in enriched soluble and membrane fractions from the control experiment were sorted into functional categories. The functional categories were adapted from the MapCave website using the TAIR10 database. Shown in Figure 6 are box and whisker plots of turnover rates of root (panel A) and shoot (panel B) proteins from the control experiment categorized by functional groups. Only proteins with at least two unique peptides were reported in Figure 6. For groups with at least three proteins, most of them had fairly similar variations in log_2_*k* values. In roots, these groups included functions such as protein synthesis, protein targeting, glycolysis, mitochondrial electron chain/ATP synthesis, cellular transport, stress response, redox reaction, and glutathione S-transferases (GSTs) metabolism (labeled in Figure as “prot.syn”, “prot.targeting”, “glycolysis”, “MC ET/ATP syn”, “transport”, “stress”, “redox”, and “GST”). In shoots, similar variations were observed in functions related to amino acid metabolism, the light reaction of photosynthesis, the Calvin cycle of photosynthesis, and protein folding (labeled in Figure as “AA met”, “PS.light”, “PS.calvin”, and “prot.folding”). These categories have well-studied proteins with known functions. Some proteins appeared to have more variation in log_2_*k* values, especially the ones in the functional categories such as redox reaction (ranging from −4.97 to −6.17 in roots, −4.48 to −6.81 in shoots), signaling (−4.89 to −6.12 in roots), development (−4.96 to −6.24 in roots), or secondary metabolism (−4.74 to −7.44 in shoots), as the proteins in these groups are involved in more varieties of function.

Some functional categories exhibited somewhat faster turnover rates, as shown by the higher median log_2_*k* values in Figure 6. It is believed that proteins with faster turnover rates could be potential control and regulation points. These include proteins involved in stress responses, signaling, protein synthesis, and protein degradation (labeled in Figure as “stress”, “signaling”, “prot.syn”, “prot.degrad”). In contrast, enzymes involved in glycolysis had the slowest turnover rates.

Overall, protein function appears to be related to turnover rates in this study. For example, in root tissues, proteins involved in cell wall formation, RNA processes, protein synthesis, hormone metabolism, and stress response (labeled in Figure as “cell wall”, “RNA”, “prot.syn”, “hormone”, and “stress”) had faster turnover rates. On the contrary, proteins involved in DNA processing, oxidative pentose phosphate pathway, major carbohydrate metabolism, and signaling (labeled in Figure as “DNA”, “OPP”, “major CHO met”, and “signaling”) had slower turnover rates. In shoot tissues, proteins related to secondary metabolism, protein degradation, and stress response (labeled in Figure as “2nd met”, “prot.degrad”, and “stress”) had higher turnover rates and appeared to turnover faster, while those involved in the Calvin cycle, hormone, and nucleotide metabolism (labeled in Figure as “PS.calvin”, “hormone”, and “nucleotide met”) had much lower turnover rates.

Some specific proteins and their turnover rates were of special interest. Table 1 and Table 2 listed the top 10 fastest and slowest proteins in the control experiment of root and shoot tissues, respectively. As listed in Appendix A, there was a 4.58-fold difference between the lowest to the highest turnover rate (*k*) among the identified root proteins (total number 221), while there was a 21.12-fold difference between the lowest to the highest turnover rate among the shoot proteins (total number 297). Therefore, the root proteome appeared to turnover faster but with less variation in general, which suggests there might be a closer correlation between regulating protein synthesis and degradation in root tissue. Stress- or redox-signaling-related proteins such as HSP 70-1 and Chaperone protein dnaJ 3 in roots or HSP 70-11 and Catalase-3 in shoots exhibited relatively rapid turnover. Proteins involved in the light reaction of photosynthesis, especially Photosystem II D2 protein and Photosystem II CP43 reaction center protein, turned over much faster than other proteins functioning in photosynthesis. Therefore, these two proteins might need to be replaced rapidly to maintain normal carbon fixation in plants. Some transport proteins such as plasma membrane ATPase 1 (AHA1) and ABC transporter G family member 36 (ABCG36; PEN3; PDR8) in the root tissue were identified as outliers in the box plot due to their extraordinarily fast turnover rates. It has been shown that the expression of *ABCG36*/*PEN3*/*PDR8* gene in seedlings is 5 to 40-fold higher than that of other ABC transporters, and its transcript abundance in leaves is comparable with the transcript levels of some housekeeping genes such as cytosolic glyceraldehyde-3-phosphate, suggesting the multiple physiological functions of ABCG36/PEN3/PDR8. It has later been reported that ABCG36/PEN3/PDR8 is an ATP-binding cassette (ABC) transporter localized on the plasma membrane and is thought to efflux indole-3-butyric acid (IBA) in root tips and several biotic and abiotic stress responses. The fast turnover rate of ABCG36/PEN3/PDR8 in seedling roots could result from the high level of protein synthesis, supporting its multiple roles in heavy metal ion tolerance as well as regulating the IBA-mediated homeostasis of auxin in roots. On the other hand, some glycosyl hydrolase family proteins, such as *beta*-glucosidase 22 (BGLU22) or *beta*-glucosidase 23 (BGLU23/PYK10), in the root or shoot tissue had the slowest turnover rates. BGLU family proteins are important for ER formation, and their hydrolytic activity for glucoside that accumulates in the roots of *Arabidopsis* has been believed to be important in defense against pests and fungi. It has been proposed that healthy seedling roots accumulate *beta*-glucosidases in the ER bodies. Therefore, when plant cells are under attack from herbivores or pathogens, *beta*-glucosidases leak from the ER body and bind to GDSL lipase-like proteins (GLLs) and Jacalin-related lectins in the cytosol to form complexes with increased enzyme activity that hydrolyze glucosides to produce toxic compounds such as scopolin. These proteins are very abundant and expressed exclusively in *Arabidopsis* seedlings, so their slowest turnover rates identified in this study suggest that BGLU22 and BGLU23 act as housekeeping proteins in *Arabidopsis* seedlings in order to rapidly trigger defense mechanisms on demand.

#### 2.3.2. Protein Function and Change in Turnover Rates Due to High Temperature

To further explore functional correlations with protein turnover changes during heat stress, the proteins with significant changes due to high temperature identified in this study were also sorted into functional categories. Figure 7A,B are box plots showing the fold changes in turnover rate in response to the higher temperature treatment (calculated from the estimated difference in log_2_*k* between the control and 30 °C using the LMM fit) across functional categories for each tissue and fraction. Only proteins with a significant change in log_2_*k* (*p* < 0.05) and at least one unique peptide in both the control and 30 °C groups were included in this analysis. In each plot, the protein categories were sorted on the *y*-axis from the largest to the smallest median difference in protein log_2_*k*. Functional categories with only one data point (one protein) were included in the plot to provide additional coverage of the functional categories. The number of proteins in each functional category is given as N along the *y*-axis of each plot. Most of the groups had median values ranging from 1.25 to 1.75 fold change. Among those identified in roots, proteins involved in redox signaling pathways, stress response, protein folding, and signaling (labeled in Figure 7 as “redox”, “stress”, “prot.folding”, and “signaling”) had the largest median changes in turnover rate. In shoots, the *beta*-glucosidase family and proteins sorted in protein folding, stress response, hormone metabolism, and secondary metabolism (labeled in Figure 7 as “prot.folding”, “stress”, “hormone met”, “2nd met”) exhibited the largest median changes in turnover rate due to heat (~1.5 fold change in *k*).

In the functional categories identified in both root and shoot soluble fractions such as redox signaling, stress response, protein degradation, and glutathione S-transferase metabolism (labeled in Figure as “redox”, “stress”, “prot.degrad”, and “GST”), shoot proteins exhibited greater changes in turnover rates than root proteins in response to heat stress, as well as secondary metabolism, protein synthesis, and stress response (labeled in Figure as “2n met”, ”prot.syn”, “stress”) in the enriched organellar and microsomal fractions (Figure 7B, C). On the other hand, proteins assigned to glycolysis, cellular transport, mitochondrial electron chain/ATP synthesis, TCA cycle, signaling, cell organization, and cell wall structure (labeled in Figure as “glycolysis”, “transport”, “MC ETC/ATP syn”, “TCA”, “signaling”, “cell”, and “cell wall”) displayed similar changes in turnover rate with heat stress in both roots and shoots, suggesting that the turnover of proteins involved in these biological processes such as mitochondrial ATP synthesis is regulated uniformly throughout the whole seedling.

Comparing the changes in turnover rates of proteins within the same functional category between different root (Figure 8A) or shoot (Figure 8B) fractions could help identify specific proteins with different or similar levels of responses to heat stress due to compartmentalization. For example, shoot proteins involved in stress responses (labeled in Figure as “stress”) appeared to be less affected by high temperature in the soluble fraction than in the membrane fractions in general. Functional categories such as the light reaction of photosynthesis, cellular transport, cell organization, mitochondrial electron transfer/ATP synthesis, protein synthesis, and glycolysis (labeled in Figure as “PS.light”, “transport”, “cell”, “MC ET/ATP syn”, “prot.syn”, and “glycolysis”) exhibited a similar breadth of responses across different fractions. This may be due to the fact that these proteins are relatively abundant, so they are being isolated in multiple fractions. Choroplastic ATP synthase subunit alpha (Atcg00120), for example, was identified in all three fractions.

## 3. Discussion

### 3.1. Heat Shock Proteins (HSPs) and Chaperones

It has long been known that the expression of stress proteins such as HSPs could be induced by heat shock at almost all stages of development, and the induction of HSPs seems to be a universal response to heat stress among organisms [41]. In the results, HSPs appear in the stress protein functional category (Figure 7 and Figure 8). While it is clear that most of the proteins listed in the table are specifically related to heat stress, such as HSP70-1, HSP70-3, HSP70-11, HSP90-2, HSP90-3, and the chaperone protein htpG family, in several of the fractions, there are additional potential stress response-related proteins predicted from the microarray gene expression data, such as RD2 protein (involved in the response to desiccation), major latex protein (MLP)-like proteins 328 and 34 (responsive to biotic stimulus), MLP-like protein 34, Dehydrin COR47 (responsive to cold), and At4g23670 protein (involved in the response to salt stress and bacterial infections). Interestingly, the root soluble fraction HSPs and stress-related proteins had smaller increases in turnover rate compared with other fractions. This significantly smaller increase in HSP and stress-related protein turnover for the root soluble protein fraction may help explain the generally much smaller change in turnover rate in that fraction compared with the other fractions.

A previous study found that stress response proteins such as heat shock chaperones and proteins associated with oxidative stress have relatively high degradation rates, although that study was performed using an enriched mitochondrial fraction of *Arabidopsis* suspension cells [42]. While it is risky to extrapolate from this prior study to intact plants, it is reasonable to postulate that the rapid turnover rate could be even more dramatic in planta. As HSPs help to prevent protein degradation, the aggregation of HSPs into a granular structure in the cytoplasm helps to protect the protein biosynthesis machinery from denaturation [7]. Our study indicates the shoot HSP90-5 (Chaperone Protein htpG family protein; At2g04030) had a 2.05-fold increase in *k* in response to heat. Overexpression of HSP90-5 in *Arabidopsis* has been shown to result in reduced plant tolerance to drought, salt, and oxidative stress, while knocking out the HSP90-5 gene results in an embryonic lethal phenotype, indicating that HSP90-5 is an essential gene [43]. It has been shown that HSP90-5 is important in maintaining the integrity of chloroplast thylakoid formation [43]. These findings, along with the dramatic change in the turnover rate of HSP90-5 when treated with high temperatures in this study, all suggest that properly controlled expression of HSP90-5 is important for plant growth and chloroplast biogenesis. HSPs such as HSP70, HSP90, and HSP60 are molecular chaperones that catalytically unfold misfolded and aggregated proteins, serving as essential cellular defenders and maintaining protein integrity. [44]. Other proteins with significant changes in turnover rate in response to high temperatures are also involved in protein degradation and protein folding functions, including several proteinases and multiple chaperones (Appendix A and Figure 7), including mitochondrial and chloroplast Chaperonin *CPN60* (HSP60) and CPN-10, which turnover rapidly in response to heat. Plastidic *CPN60 alpha* and *beta* are crucial for plastid division in *Arabidopsis,* ensuring the proper folding of stromal plastid division proteins and being essential for chloroplast development [45]. The observed change in *CPN60* turnover rates is somewhat correlated to the study revealing the slightly reduced expression of CPN-60 in seedling shoots when encountering the elevated temperature at 28 °C [45]. Another chaperone protein, AtBAG7 (At5g62390), exhibited a faster turnover rate at elevated temperatures. AtBAG7 is required to maintain the c and is localized in the endoplasmic reticulum, which is unique among BAG family members [19]. It has been proposed that activity may be regulated post-translationally, given that its gene expression does not appear to be affected by heat or cold stresses [19]. Since AtBAG7 directly interacts with an HSP70 paralog, AtBAG7 activity is likely regulated post-translationally through modulation of protein turnover [19].

### 3.2. Photosynthesis and Carbon Assimilation

As temperature is a crucial factor affecting photosynthetic activity in plants, as expected, proteins involved in photosynthesis, including components of photosystems I and II (PSI and PSII), the cytochrome b6-f complex, chloroplast ATP synthase, and the Calvin cycle, were identified as having varying degrees of change in turnover in response to heat. Prior heat stress-related studies found that the oxygen-evolution complex (OEC) of PSII is the main target of heat stress [46]. From this study, changes in turnover rates of OEC subunits were around 1.21–1.42 fold, similar to the majority of the proteins involved in photosynthesis, in response to heat (Appendix A). There were extreme cases such as RuBisCO activase (At2g39730) and chlorophyll a/b binding protein (LHCB6; At1g15820) that exhibited larger 1.57 and 1.60 fold changes in *k*, respectively. As it is highly sensitive to heat denaturation, RubisCo activase is thought to be a key element involved in mediating the heat-dependent regulation of carbon assimilation, as it could limit the photosynthetic potential of plant tissues at high temperatures [14]. Although the enzyme activity of RubisCo activase was not decreased until the temperature was higher than 37 °C in cotton and tomato leaves [14], our study suggests that this enzyme in *Arabidopsis* seedlings could “sense” relatively mild elevated temperatures such as 30 °C in terms of protein turnover. It is hard to judge from the results whether the turnover rates of proteins in PSII and light-harvesting complex II (LHCII) were more affected by high temperatures than PSI, as it has long been believed that PSII is more vulnerable to elevated temperatures [47,48]. A comparison, however, of the differences between PSI and PSII protein turnover following heat stress should indicate the relative heat tolerance of the two photosystems under mildly elevated temperature conditions. To this end, LHCB6, which is associated with PSII turns over significantly faster (1.60 fold change in *k*) after heat treatment than the Photosystem I reaction center subunit III (1.25 fold change in *k*). Notably, these rate changes are on the high and low extremes of the range of changes observed for protein components of photosynthesis. LHCB6 is a monomeric antenna protein of PSII, that participates in zeaxanthin-dependent photoprotective mechanisms and is therefore thought to be specialized in enhancing photoprotection under excess light conditions. The presence of the protein is often associated with the adaptation of plants to terrestrial ecosystems [49]. Heat stress at temperatures around 38–40 °C has been demonstrated to cause structural changes in the thylakoid membranes, as well as increased phosphorylation of LHCIIs and PSII core subunits, migration of phosphorylated LHCII from the grana stacks to the stroma lamellae, and cyclic electron flow within PSI [50]. It will be interesting to study if the change in LHCB6 turnover could be related to the above observations at 40 °C, even when mild temperature conditions such as 30 °C are employed.

### 3.3. Redox Homeostasis: HSPs, Catalases, and Peroxidases

The turnover rates of proteins involved in the production of reactive oxygen species (ROS) were also affected by high temperatures. These include several different types of HSPs, catalases, and peroxidases. An additional group of antioxidant enzymes, including GST, DHAR, and thioredoxins, exhibited significant heat-related changes in turnover (Appendix A). Among those, GST class Tau-member 19 (GSTU19; At1g78380), the most abundant GST in *Arabidopsis*, exhibited the smallest difference in turnover rate (1.31-fold change) in roots but showed a much larger difference (1.75-fold change) in turnover rate in shoots.

Hydrogen peroxide (H_2_O_2_) is an important signaling molecule in plant environmental responses, and heat shock-induced H_2_O_2_ accumulation is required for efficiently inducing the expression of small HSP and ascorbate peroxidase genes (*APX1* and *APX2*) [51]. Among several types of H_2_O_2_-metabolizing proteins, catalases are highly active enzymes that do not require cellular reductants as they catalyze the dismutation reaction of two molecules of H_2_O_2_ to generate one molecule of O_2_ and two of H_2_O. A 1.40-fold change in turnover rate *k* was observed for catalase-3 (CAT3; At1g20620) in shoots upon temperature elevation. APXs are also known to be important H_2_O_2_-scavenging enzymes, but they use ascorbate as an electron donor. Their function is tightly linked to ROS signaling pathways and the regulation of cellular ROS levels [51]. In this study, there was a moderate increase in APX1 (At1g07890) turnover rates under heat stress conditions in both root and shoot tissues. *APX1* is expressed in roots, leaves, stems, and many other tissues [52], and mutation in *Arabidopsis APX1* exhibits increased accumulation of cellular H_2_O_2_ and suppresses growth and development [53]. It has been reported that APX1 activity could be partially inhibited in roots through modification by S-denitrosylation in an auxin-dependent manner [54]. APX1 could be an interesting research target to explore the links between nitric oxide (NO), H_2_O_2_, auxin hormone signaling, and heat stress.

### 3.4. Special Cases: Decreases or Major Increases in Protein Turnover Rates in Response to Heat Stress

#### 3.4.1. GDSL Esterase/Lipase Family

GDSL esterase/lipase 22 (GLL22; At1g54000) showed slightly reduced turnover rates in both root organellar and microsomal fractions (fold change in *k* of about 0.86 and 0.89, respectively), indicating that GLL22 becomes more stable and/or has reduced transcription or translation when transferred to 30 °C. It has been proposed that under pathogen or herbivore attack, GLL22 may aggregate with *beta*-glucosidases (BGLU 21, 22, and 23) and other Jacalin-related lectins (JALs) in the cytosol [55]. It is possible that under temperature stress, GLL22 turns over slower due to being recruited into more stable complexes. The change in turnover rates of the BGLU protein family, on the other hand, had a wide variation across root or shoot protein fractions (1.09~2.14 fold change due to heat). BGLU proteins appeared to turn over faster in shoots than in roots; thus, the turnover rates of BGLUs in shoots could be more affected by heat stress than BGLUs in roots. Similar results were observed for JAL proteins such as Jacalin-related lectin 30 (PYK10-binding protein 1; At3g16420), Jacalin-related lectin 33 (JAL 33; At3g16450), and Jacalin-related lectin 34 (JAL 34; At3g16460), whose turnover rates also had a greater change in shoots than roots when under heat stress, suggesting these stress-responsive proteins in shoots may be compromised when plants encounter heat stress.

#### 3.4.2. 14-3-3 and V-, P-Type ATPase

It is intriguing to observe signaling proteins such as 14-3-3 family proteins and proton pump v- and p-type H+-ATPases with significant changes in turnover rate due to elevated temperature because of their known roles in ABA signaling in response to abiotic stress. Increased H_2_O_2_ production under multiple different abiotic stress conditions has been shown to result in elevated levels of ABA, which may in turn be involved in the induction of the temperature stress response in plants [17]. Plant 14-3-3 family proteins function in a wide range of cellular processes. Two 14-3-3 proteins show fairly large changes in protein turnover in response to heat stress: 14-3-3-like Protein GF14 mu (General regulatory factor 9; At2g42590), and 14-3-3-like Protein GF14 epsilon (General regulatory factor 10; At1g22300), with 1.61 and 1.45 fold changes, respectively. It has been discovered that 14-3-3 mu participates in light sensing during early development through phytochrome B signaling and affects the time of transition to flowering via interaction with CONSTANS [56]. As T-DNA mutants of the 14-3-3 mu gene exhibit shorter root lengths and a dramatic increase in the numbers of chloroplasts in the roots [57], it is possible that the difference in heat stress response between root and shoot tissues is related to its role in chloroplast development. On the other hand, the 14-3-3 epsilon protein may be involved in brassinosteroid (BR) signaling, like the 14-3-3 lambda protein, as the 14-3-3 epsilon protein has been shown to interact with the BZR1 transcription factor in a yeast-two hybrid screen [58]. Therefore, these proteins involved in signal transduction may be affected by heat stress, thus influencing BR hormone regulation.

### 3.5. Stability of Proteins Involved in Primary Metabolism and Energy Production

Interestingly, proteins involved in glycolysis, the TCA cycle, mitochondrial electron transport, and ATP synthesis showed relatively smaller changes in turnover rates under heat stress across both root and shoot tissues. This observation suggests that maintaining stability and homeostasis in these primary metabolic pathways and energy production processes is crucial for plant survival under stress conditions. Glycolysis and the TCA cycle are central metabolic pathways that provide energy, reducing power, and precursors for various biosynthetic processes [59]. Similarly, mitochondrial electron transport and ATP synthesis are essential for generating the energy currency required for cellular functions [60]. Any significant perturbations in these pathways could lead to energy depletion, metabolic imbalances, and ultimately, cell death [61,62].

The relatively stable turnover rates of proteins involved in these pathways under heat stress may indicate the presence of protective mechanisms that confer higher stability to these protein groups compared to others. One potential mechanism could be the intrinsic thermostability of the enzymes involved in these pathways. Many glycolytic enzymes, such as glyceraldehyde-3-phosphate dehydrogenase and enolase, have been shown to possess high intrinsic thermostability [63]. Similarly, enzymes of the TCA cycle, such as citrate synthase and malate dehydrogenase, are known to be stable at elevated temperatures [64]. The thermostability of these enzymes may be attributed to their unique structural features, such as a higher number of salt bridges, hydrogen bonds, and hydrophobic interactions [65].

Another potential mechanism for maintaining stability could be the presence of chaperones and HSPs that specifically target and protect these metabolic enzymes during heat stress. HSPs have been shown to interact with and stabilize various enzymes involved in primary metabolism and energy production [66]. For instance, the chaperone HSP90 has been found to interact with and stabilize the mitochondrial ATP synthase complex under heat stress [67]. Additionally, post-translational modifications, such as phosphorylation and acetylation, may play a role in enhancing the stability of these proteins under stress conditions [68]. The maintenance of stable turnover rates for proteins involved in primary metabolism and energy production during heat stress highlights their importance in plant survival and adaptation. Further research into the specific mechanisms conferring stability to these protein groups could provide valuable insights into the development of stress-resilient crops.

### 3.6. Differential Responses of Root and Shoot Proteomes to Heat Stress

The results of this study indicate that the shoot proteome exhibits a greater change in protein turnover rates in response to elevated temperatures compared to the root proteome. In plants, it is believed that root growth is more sensitive to acute heat stress than shoot growth, as the high soil temperature is more detrimental than the high air temperature, and a lower soil temperature could help plants survive when grown under high air temperatures [69]. One possible explanation for this differential response could be the direct exposure of shoots to high temperatures and light intensity, which can lead to increased production of ROS and oxidative stress [70]. In contrast, roots are buffered from extreme temperature fluctuations by the soil environment [71]. Additionally, shoots contain photosynthetic machinery, which is highly sensitive to heat stress [72]. The faster turnover of proteins involved in redox signaling, stress response, and photorespiration in shoots may represent an adaptive mechanism to maintain cellular homeostasis and protect the photosynthetic apparatus under elevated temperatures [73]. These findings are consistent with previous studies that have reported tissue-specific responses to abiotic stress in plants. For instance, a study by Liu et al. [74] found that the shoot proteome of rice exhibited more significant changes than the root proteome under drought stress. Similarly, a study by Ghosh et al. reported that the shoot proteome of wheat showed a greater response to heat stress compared to the root proteome [75]. The differential responses of root and shoot proteomes to abiotic stress may reflect the distinct physiological roles and adaptations of these tissues in stress tolerance [76]. Further research is needed to elucidate the molecular mechanisms underlying these tissue-specific responses and their implications for plant stress resilience.

Taking the fast turnover rate of plasma membrane proton pump (ATPase 1) in roots under controlled temperature, for example (Table 1), the establishment of protein machinery for metabolite uptake could be essential for growth at this stage. Although several proteins had dramatically long half-lives (Table 1 and Table 2), the average protein turnover rates measured in this study were much faster than the average protein turnover rates in 21 to 26-day-old adult *Arabidopsis* leaves (approximately 4.6 days), as reported in the unpublished work from Millar et al. (presented at the 2015 ASPB conference), suggesting that more rapid protein turnover may be required in the seedling stage than the adult stage in plants.

### 3.7. Expanding upon Prior 15N-Labeling Studies: Progress and Limitations in the Current Study

Despite continuous advancements in liquid chromatography (LC)-coupled mass spectrometry (MS) instrumentation over the past two decades, a limited number of studies have explored the effects of stress conditions on protein dynamics or turnover [77]. One close example is documented by Li et al. [78], who utilized ^15^N-labeling and two-dimensional fluorescence difference gel electrophoresis with LC-MS/MS to measure the protein degradation rates of 84 proteins in *Arabidopsis* suspension cells. They subsequently calculated protein synthesis rates based on degradation rates and changes in protein relative abundance. The study concluded that protein turnover rates are generally correlated with protein function and among protein complex subunits. Using a similar approach, Nelson et al. measured the degradation rate of mitochondrial proteins using *Arabidopsis* cell cultures with ^15^N-label incorporation at different time points [42]. These studies employed the *Isodist* algorithm [79] to assign the isotopic abundance to natural abundance and labeled peptide mass spectral data to obtain Relative Isotope Abundance (RIA) values for each peptide throughout the time course. However, a limitation of these approaches is the loss of individual peptide contributions to overall protein turnover due to the use of median peptide RIA values for each protein. In order to detect significant changes in protein turnover rates across different treatments, such as stress conditions, the individual contributions of specific peptides to the overall protein turnover may be easily lost due to the use of median peptide RIA values for each protein. This unnecessarily discards potentially important information regarding the inherent heterogeneity of intracellular protein populations.

In addition to the studies conducted in plants, proteome-scale analysis has been demonstrated in barley leaves. This was achieved through gas chromatography-mass spectrometry analysis of free amino acids and LC-MS analysis of proteins, enabling the tracking of the enrichment of ^15^N into the amino acid pools [80]. Another study utilized a ^13^CO_2_-labeling approach to quantify the synthesis and degradation rates of selected proteins in Arabidopsis adult plants [81]. Similarly, the degradation rates of ~1200 Arabidopsis leaf proteins have been characterized at different growth and development stages, revealing that protein complex membership and specific protein domains can serve as predictors of degradation rate [82]. However, a comprehensive analysis of plant proteomes under stress conditions has been limited, potentially due to challenges in implementing the methodological pipeline, from processing MS data to generating turnover rate calculations.

The experimental approach used in this study, combining ^15^N-labeling, UHPLC-HRMS/MS analysis, and the *ProteinTurnover* algorithm, provides a powerful tool for investigating protein turnover dynamics in plants under stress conditions. The ^15^N-labeling method allows for accurate quantification of protein synthesis and degradation rates, while the UHPLC-HRMS/MS analysis enables high-throughput identification and quantification of proteins [82]. The *ProteinTurnover* algorithm streamlines the data processing and calculation of protein turnover rates, making it easier to analyze large proteomic datasets [36]. However, there are some limitations to this approach. For instance, the detection of low-abundance proteins may be limited by the sensitivity of the mass spectrometer [83]. Additionally, the accuracy of protein turnover rate calculations may be affected by factors such as incomplete ^15^N incorporation or protein degradation during sample preparation [84].

### 3.8. Summary and Future Research Directions

In this study, an elevated temperature of 30 °C was applied for durations ranging from 8 to 48 h. Despite being relatively moderate compared to typical heat stress studies, it has been demonstrated that even a modest change in temperature, such as transferring 12-day-old *Arabidopsis* seedlings from 12 to 27 °C for 2 h, can significantly alter the expression of over 5000 genes by at least 2-fold [85]. The present study’s moderate heat treatment aligns with the moderately elevated temperature, contrasting with the heat stress conditions in the Mittler study [85]. This suggests that different heat sensors and signaling pathways may perceive these temperature regimes differently.

The results of this study suggest that heat stress causes a greater change in the shoot proteome than the root proteome. The analysis found that proteins involved in redox signaling, stress response, protein folding, and secondary metabolism had the most significant turnover rate changes under heat stress, especially in shoot tissues. Significant changes in the turnover rates of HSP70, HSP90, and the chaperone protein htpG underscore their protective roles against heat-induced protein degradation. Specifically, HSP90-5 is crucial for maintaining chloroplast integrity under stress. RuBisCO activase exhibited increased turnover, suggesting limitations in photosynthetic efficiency at higher temperatures. Proteins such as GSTs, catalases, and peroxidases involved in redox homeostasis showed diverse responses, indicating their roles in oxidative stress management. Notable changes in proteins such as GDSL esterase/lipase and 14-3-3 family proteins highlight their involvement in broader stress responses and signaling pathways related to abscisic acid and brassinosteroids. In contrast, proteins involved in primary metabolism and energy production, such as glycolysis, the TCA cycle, mitochondrial electron transport, and ATP synthesis, showed smaller turnover rate changes under heat stress in both root and shoot tissues. This study highlights the importance of protein turnover dynamics in plant stress adaptation, showing how changes in protein synthesis and degradation rates help plants survive at high temperatures.

Future studies could address the limitations of this study by using more sensitive mass spectrometry techniques, such as targeted proteomics or data-independent acquisition [86], and by optimizing sample preparation protocols to minimize protein degradation. Complementary techniques, such as pulse-chase labeling or single-cell proteomics [87], could also be used to validate and extend the findings of this study. In addition, future research could investigate the turnover dynamics of low-abundance proteins, which may play important regulatory roles in stress response [88]. Examining the effects of different stress durations or intensities on protein turnover could provide insights into the temporal dynamics and dose-response relationships of stress-induced changes in protein metabolism [89]. Furthermore, future studies could focus on understanding the mechanisms behind the differential proteomic responses to moderate heat stress, particularly comparing resilience mechanisms in roots versus shoots. Employing advanced techniques such as metabolic flux analysis [90,91,92] would be key for quantifying metabolic changes and linking them to specific biochemical pathways, enhancing our understanding of plant metabolic adjustments to heat stress [93,94,95]. Finally, integrating systems biology approaches to correlate transcriptomic, proteomic, and metabolomic data could provide comprehensive insights into the regulatory networks and pathways activated during heat stress, potentially revealing critical regulatory nodes for enhancing plant heat tolerance [96,97,98].

## 4. Materials and Methods

### 4.1. Materials

Distilled, deionized water was prepared with a Barnstead B-pure water system (Thermo Scientific, Waltham, MA, USA). Acetonitrile (CHROMASOLV^®^ Plus for HPLC, ≥99.9%), formic acid (ACS reagent ≥ 96%), and acetone (CHROMASOLV^®^ Plus for HPLC, ≥99.9%) were obtained from Sigma-Aldrich (St. Louis, MO, USA). Triton X-100 was obtained from ICN Biochemicals Inc. (Solon, OH, USA). A 99 atom% of K^15^NO_3_ and 98 atom% of Ca(^15^NO_3_)_2_ were obtained from Cambridge Isotopes Laboratories, Inc. (Andover, MA, USA). Sequencing grade modified trypsin was purchased from Promega (Madison, WI, USA). Pierce C18 Spin columns were obtained from Thermo Scientific (Pierce Biotechnology, Thermo Scientific, Rockford, IL, USA). Micro-centrifuge tubes used for the proteomics study in this thesis were “Protein LoBind Tube 1.5 mL”, obtained from Eppendorf AG (Hamburg, Germany). Nylon filter membranes (mesh opening 20 μm, Cat. #146510) were obtained from Spectrum Laboratories Inc. (Rancho Dominguez, CA, USA).

### 4.2. Plant Growth and Labeling Conditions

All experiments were conducted using *Arabidopsis thaliana* ecotype Columbia Col-0. Before germinating on a nylon filter membrane placed on the top of ATS agar plates, seeds were sterilized with 30% (*v*/*v*) bleach containing 0.1% (*v*/*v*) Triton X-100 and vernalized at 4 °C for two days. The seedlings were then grown under continuous fluorescent light (~80 μmole photon m^−2^ s^−1^) at 22 °C for 8 days. For the heat-treated group, these 8-day-old seedlings along with the nylon membrane (mesh opening 20 μm, Cat. #146510, Spectrum Laboratories Inc., Rancho Dominguez, CA, USA) were then transferred onto fresh ATS [99] media containing 99 atom% K^15^NO_3_ and 98 atom% Ca(^15^NO_3_)_2_ (Cambridge Isotopes Laboratories, Inc., Andover, MA, USA) (^15^N-medium) and then transferred to the 30 °C growth chamber. For the control group, seedlings were continuously grown at 22 °C after being transferred to the ATS medium with the normal nitrogen source (^14^N-medium).

For both the control and high-temperature groups, crude proteins were extracted at 0, 8, 24, 32, and 48 h after ^15^N incorporation (time 0 samples were shared by both groups). Prior to transferring seedlings from ^14^N- to ^15^N-media, the ATS liquid medium lacking K^15^NO_3_ or Ca(^15^NO_3_)_2_ was used to rinse the seedlings.

### 4.3. Proteomic Sample Preparation

For the proteomic analysis of *Arabidopsis* seedlings, hypocotyl and cotyledons (as “shoot” samples) were dissected from root tissues. From root and shoot tissues, soluble and membrane proteins were extracted and enriched by differential centrifugation, as described previously by Fan et al. [36] in the stable isotope incorporation experiments. The proteolysis of soluble protein, membranous protein fractions derived from 1500× *g* (organelle), and 100,000× *g* (microsomal) pellets were processed as described previously [36]. The resulting peptides obtained from soluble or membrane protein fractions were purified by C18 solid phase extraction using the C18 Spin column (Pierce Biotechnology, Thermo Scientific, Rockford, IL, USA) and per the manufacturer’s protocol. After purification, peptides were concentrated under vacuum to dryness using a SpeedVac concentrator (Savant) and were re-suspended in 5% (*v*/*v*) acetonitrile, 0.1% formic acid prior to UHPLC-HRMS/MS analysis.

### 4.4. UHPLC-HRMS/MS Analysis

The tryptic peptides were analyzed by UHPLC-HRMS/MS using a Q Exactive hybrid quadrupole orbitrap mass spectrometer with an Ultimate 3000 UHPLC inlet (Thermo Fisher Scientific, Vacaville, CA, USA) equipped with an ACQUITY UPLC BEH C18 reversed-phase column (Waters, 2.1 mm × 100 mm, 1.7 µm particle size). Solvent A (0.1% (*v*/*v*) formic acid in H_2_O) and B (0.1% (*v*/*v*) formic acid in acetonitrile) were used as mobile phases for gradient separation. The UHPLC-HRMS/MS analysis method, which involves the separation of tryptic peptides using a UHPLC system equipped with an ACQUITY UPLC BEH C18 reversed-phase column and their subsequent detection and identification using a Q Exactive hybrid quadrupole orbitrap mass spectrometer, has been previously described by Fan et al. [36].

### 4.5. Protein Identification

The protein identification process, which involves converting .raw files to mzXML and mgf formats, database searching using OMSSA against the UniProt *Arabidopsis thaliana* database combined with the cRAP database, and validation of MS/MS-based peptide and protein identifications using Scaffold, has been previously described by Fan et al. [36].

### 4.6. Calculation of Protein Turnover Rates

The workflow for using the *ProteinTurnover* algorithm is described in the following steps: (1) Data preparation. The Scaffold spectrum report (CSV format) and all MS data (mzXML format) were uploaded for access by the R script; (2) Parameter settings. Parameters such as stable isotope (^15^N) used for labeling, experimental design (incorporation), peptide ID confidence threshold (80), spectral fitting model (beta-binomial), and nonlinear regression setting (log_2_*k*) were defined; (3) Outputs generated. After finishing the analysis of a dataset, the results were compiled in a summary HTML file, which includes model plots (spectral fitting by MLE), EIC plots, and regression plots (relative abundance fits) for each individual peptide to be used as needed for manual inspection. The *ProteinTurnover* R script also generates a spreadsheet (.csv) containing peptide turnover information, which includes the peptide amino acid sequences, protein UniProt accession numbers (ID), visual scores, log_2_*k* values, and standard errors of log_2_*k*.

The calculation of the log_2_ value for each turnover rate constant (*k*) of each peptide in isotope label incorporation experiments, which involves performing a non-linear regression of the distribution abundance ratios of the unlabeled peptide population against time assuming a single exponential decay, has been previously described by Fan et al. using the *ProteinTurnover* algorithm [36].

Protein turnover typically exhibits first-order kinetics, and the first-order rate constant (*k*) is related to the half-life of the particular peptide by the expression t_1/2_ = (ln(2))/*k*. In this study, the turnover rate was represented by the log_2_*k* values, which are more normally distributed than the untransformed rate constants. After obtaining the turnover results from *ProteinTurnover*, peptides were selected for subsequent inclusion in protein turnover calculations by applying the following filtering criteria: (1) the visual score of the spectral fitting to the beta-binomial model must be >80; (2) the standard error of the turnover rate must be <10; and (3) data must be available for three or more time points. The log_2_*k* data of the selected and unique peptides were averaged to obtain the protein turnover rate.

### 4.7. Estimating the Difference in Log_2_k Due to Heat Stress

The selected peptides were analyzed in R to calculate the difference in turnover rate between the control and treated groups. A linear mixed model (LMM) fit with restricted maximum likelihood (using the lme4 package in R) was applied to estimate the change of protein log_2_*k* between the control and heat-treated groups based on peptide log_2_*k* data. The used formula is listed as follows:log_2_*k*~0 + ID + ID:temp + (1|Sequence:ID),
where “ID” represents the protein UniProt accession number, “temp” represents either the control or 30 °C group, and “Sequence” represents the peptide amino acid sequence. In the end, only proteins with significant changes in log_2_*k* (*p*-value less than 0.05) were included in Appendix A. Only proteins with more than one computable unique peptide in both the control and heat-treated groups were selected to generate histograms and box plots (Figure 5, Figure 6 and Figure 7).

## 5. Conclusions

This study provides a comprehensive overview of the dynamics of proteins in plants in response to moderate heat stress. Conducted at the cellular level, it involved the separation of soluble and membrane enrichments using ^15^N-stable isotope labeling and the *ProteinTurnover* algorithm for automated data extraction and turnover rate calculation. A total of 571 proteins with significant changes in turnover rates were identified in response to elevated temperatures in Arabidopsis seedling tissues. Root proteins involved in the redox signaling pathway, stress response, amino acid metabolism, GST metabolism, protein synthesis, protein degradation, and cellular organization exhibited a less pronounced change in turnover than shoot proteins. Conversely, proteins involved in GST metabolism, photorespiration, protein folding, secondary metabolism, stress response, redox signaling pathway, and the *beta*-glucosidase family displayed the most notable alterations in turnover rates under elevated temperature conditions. Notably, proteins involved in major carbohydrate metabolism, glycolysis, protein synthesis, and mitochondrial ATP synthesis showed the smallest changes in turnover under this stress. This comprehensive study underscores the adaptive mechanisms of plants at the proteomic level in response to heat stress, offering insights for future agricultural strategies aimed at enhancing crop resilience and productivity in the context of global climate change.

## Figures and Tables

**Figure 1 ijms-25-05882-f001:**
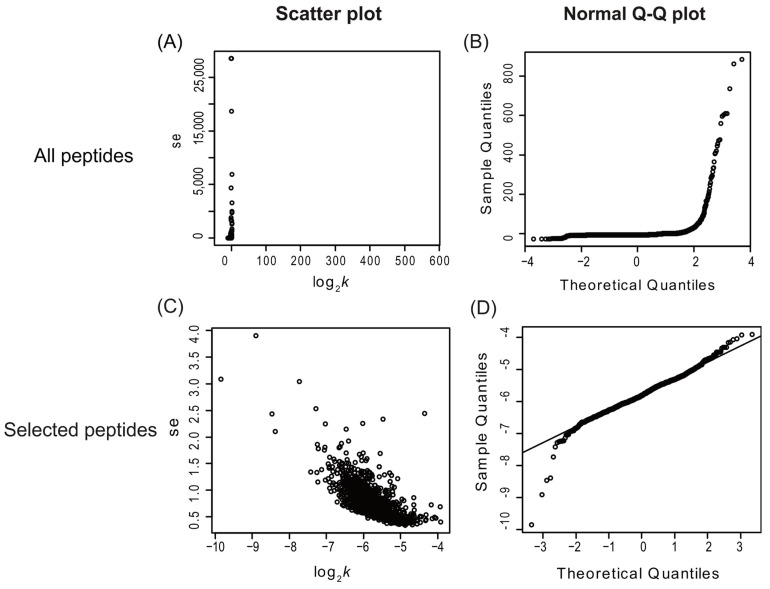
Scatter plots and normal Q-Q plots of all identified *Arabidopsis* peptides (top) vs. peptides selected with visual scores higher than 80, standard error lower than 10, and identified in at least three labeling time points (bottom). The panels on the left (**A**,**C**) are scatter plots of the standard error of log_2_*k* (se.log_2_*k*) against log_2_*k*; the panels on the right (**B**,**D**) are normal Q-Q plots of each peptide’s turnover rate (log_2_*k* values). This figure shows only the peptide data from the enriched shoot soluble fraction and includes data combined from both the control and heat treatment groups. The number of peptides is 10,400 (**A**,**B**) and 1273 (**C**,**D**). se, standard error.

**Figure 2 ijms-25-05882-f002:**
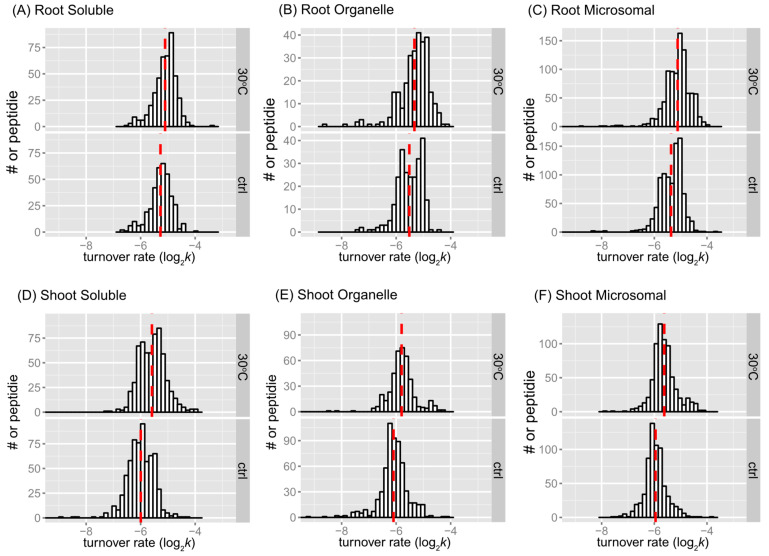
Peptide turnover rate distributions by tissue, fraction, and treatment. Histograms show peptide log_2_*k* values plotted for enriched soluble, organelle, and microsomal fractions of root (**A**–**C**) or shoot (**D**–**F**) tissues. The control (ctrl) and 30 °C groups are plotted in the bottom and top frame, respectively. The *y*-axis is the number of peptide counts. The mean value is plotted as the dashed line in red. The bin width is 0.15 for all histograms.

**Figure 3 ijms-25-05882-f003:**
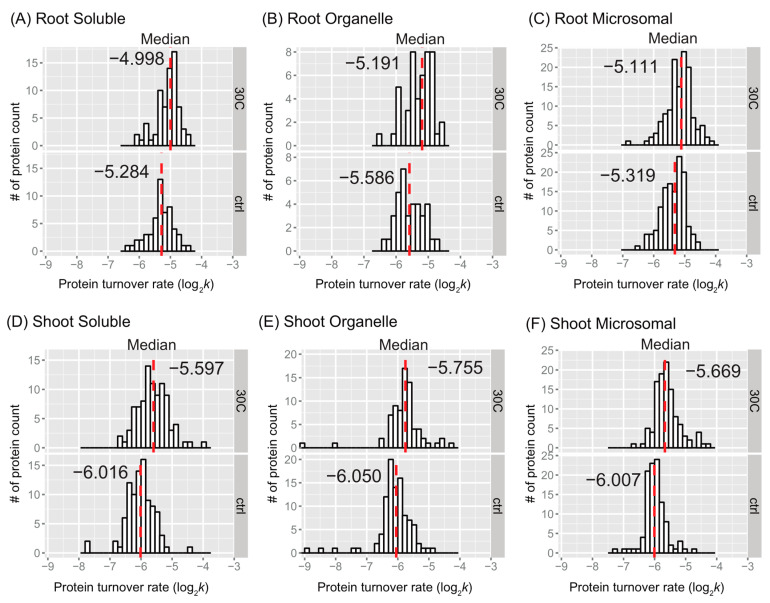
Protein turnover rate distributions by tissue, fraction, and treatment. Histograms show protein log_2_*k* values plotted for enriched soluble, organelle, and microsomal fractions of root (**A**–**C**) or shoot (**D**–**F**) tissues. The control (ctrl) and 30 °C groups are plotted in the bottom and top frames, respectively. The *y*-axis is the number of protein counts. The median value is labeled and plotted as the dashed line in red. The bin width is 0.15 for all histograms.

**Figure 4 ijms-25-05882-f004:**
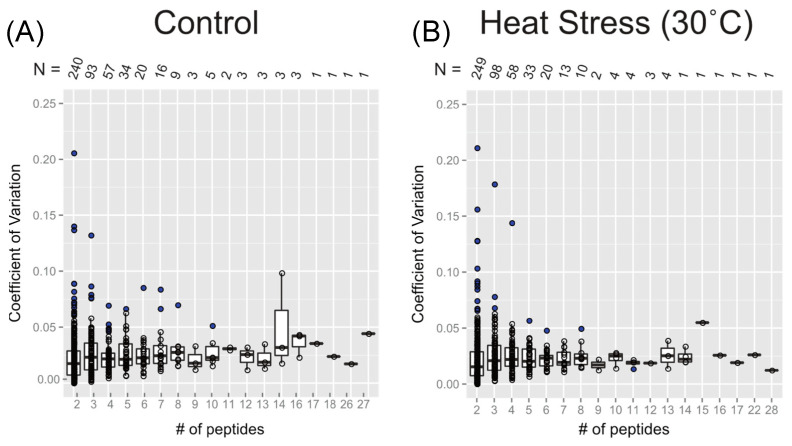
Box plots of the coefficient of variation (CV) of protein turnover rates plotted as a function of the number of peptide rates used in each calculation. The value of CV was calculated from the standard deviation of log_2_*k* divided by the mean of log_2_*k*. The dataset used in this plot analysis compromises both unique and shared peptides, separated according to the treatment groups: the control temperature (**A**) and the elevated temperature of 30 °C (**B**). Boxes show the interquartile range (IQR) of turnover rates of proteins. The error bar represents the entire range of rates, and the blue dots represent outliers (1.5 IQR). The number of data points in each *x*-axis category is given as *N*, below the *x*-axis of both plots.

**Figure 5 ijms-25-05882-f005:**
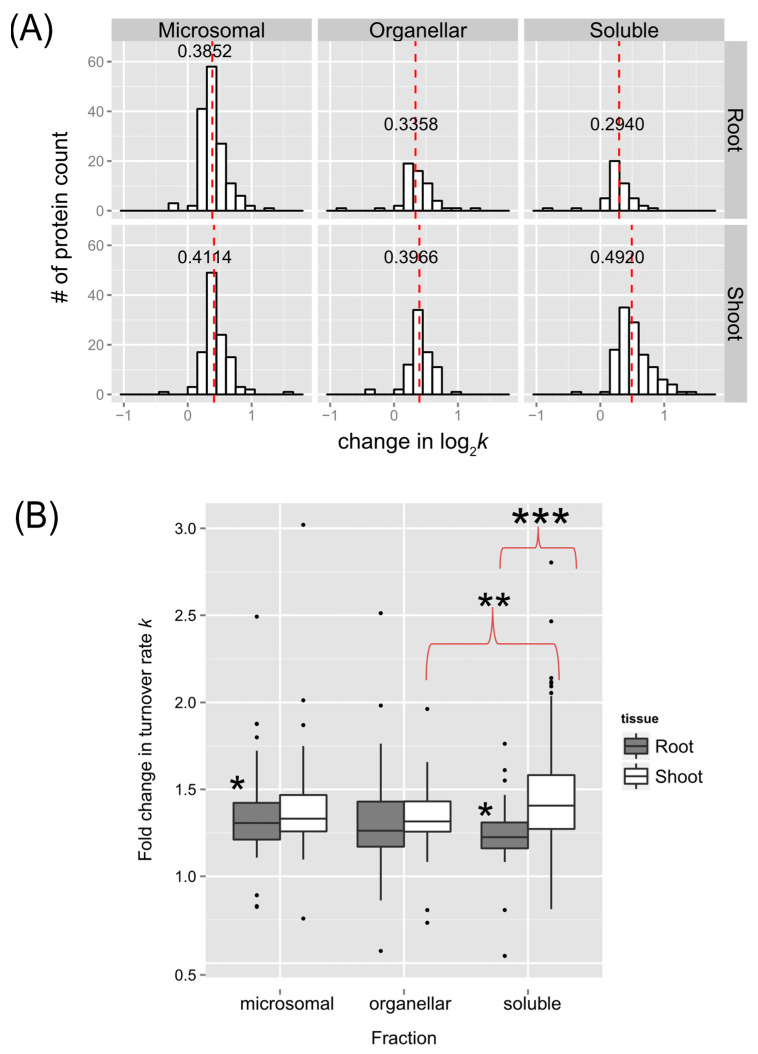
The distribution of changes in protein turnover rates across different tissues and enriched fractions in response to elevated temperature. (**A**) Histograms depict distributions of estimated changes in protein turnover rate (log_2_*k*) in response to 30 °C, plotted for the soluble, organelle, and microsomal fractions of roots (top panel) or shoots (bottom panel), respectively. The bin width is 0.15 for all histograms. The median value is labeled and plotted as the dashed line in red. (**B**) Box plots of the estimated fold change in protein turnover rate constant (*k*) in response to 30 °C of proteins identified in the root and shoot enriched soluble, organelle, and microsomal fractions. The analyzed data include only proteins with a significant change in log_2_*k* (*p* < 0.05) and at least one unique peptide identified in both the control and 30 °C groups, which was estimated using an LMM approach after peptide selection criteria were applied. Boxes show the interquartile range (IQR) of change in turnover rates *k*. The error bar represents the entire range of rates, and the closed circles represent outliers (1.5 IQR). The estimated changes in turnover rates were analyzed by Tukey’s HSD (Honest Significant Difference) test, with * for *p* < 0.05, ** for *p* < 0.01, and *** for *p* < 0.001.

**Figure 6 ijms-25-05882-f006:**
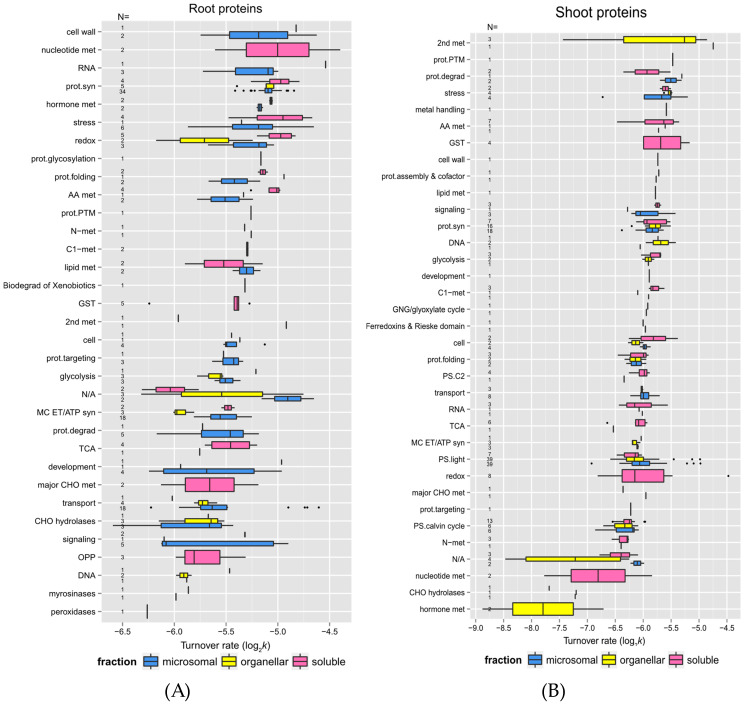
The relationship between protein function and protein turnover rates. Box plots of protein turnover rate log_2_*k* for root (**A**) and shoot (**B**) proteins from the control experiment are sorted by functional categorization, adapted from the MapCave website (http://mapman.gabipd.org/web/guest/mapcave, accessed on 24 September 2014) using the TAIR10 database. Outliers are shown as closed circles. The used data include only proteins with at least two unique peptides. The number of proteins in each function category is given as *N* along the *y*-axis of both plots. The protein count of each function group is also labeled in the plot. Abbreviations: 2nd met, secondary metabolism; AA met, amino acid metabolism, C1-met, single carbon metabolism; cell, cell organization; cell wall, cell wall formation; CHO hydrolases, miscellaneous gluco-, galacto- and mannosidases; DNA, DNA processing; Glc-, Gal- and mannosidases, glucosyl-, galactosyl- and mannosyl- glycohydrolases; GNG, gluconeogenesis; GST, glutathione S-transferase metabolism; hormone met, hormone metabolism; lipid met, lipid metabolism; major CHO met, major carbohydrate metabolism; MIP, major-intrinsic proteins; MC ET/ATP syn, mitochondrial electron transport/ATP synthesis; N-met, nitrogen metabolism; OPP, oxidative pentose phosphate pathway; prot.assembly, protein assembly and cofactor ligation; prot.degrad, protein degradation; prot.folding; protein folding; prot.targeting, protein targeting; prot.PTM, protein post-translational modification; prot.syn, protein synthesis; PS.C2, photorespiration; PS.light, the light reaction of photosynthesis; PS.calvin cycle; the Calvin Cyle of photosynthesis; RNA, RNA processing; S-assimilation, sulfur assimilation; stress, stress responses; TCA, tricarboxylic acid cycle; transport, cellular transport; N/A, protein function not assigned.

**Figure 7 ijms-25-05882-f007:**
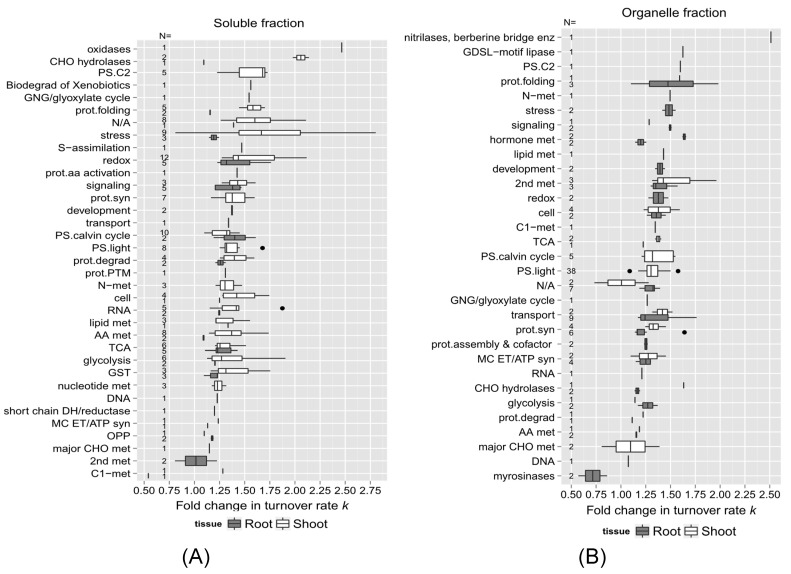
The relationship between protein function and change in turnover due to elevated temperature. Box plots of the estimated fold change in protein turnover rate constant (*k*) in response to 30 °C for root and shoot proteins are sorted by functional categorization, adapted from the MapCave website using the TAIR10 database. Outliers are shown as closed circles. (**A**) Enriched soluble fraction of root or shoot tissue homogenate. (**B**) Enriched organelle fraction of root or shoot tissue homogenate. (**C**) Enriched microsomal protein fraction of root or shoot tissue homogenate. The data include only proteins with a significant change in log_2_*k* (*p* < 0.05) and at least one unique peptide identified in both the control and 30 °C groups. The number of proteins in each function category is given as *N* along the *y*-axis of all plots. N/A, protein function not assigned.

**Figure 8 ijms-25-05882-f008:**
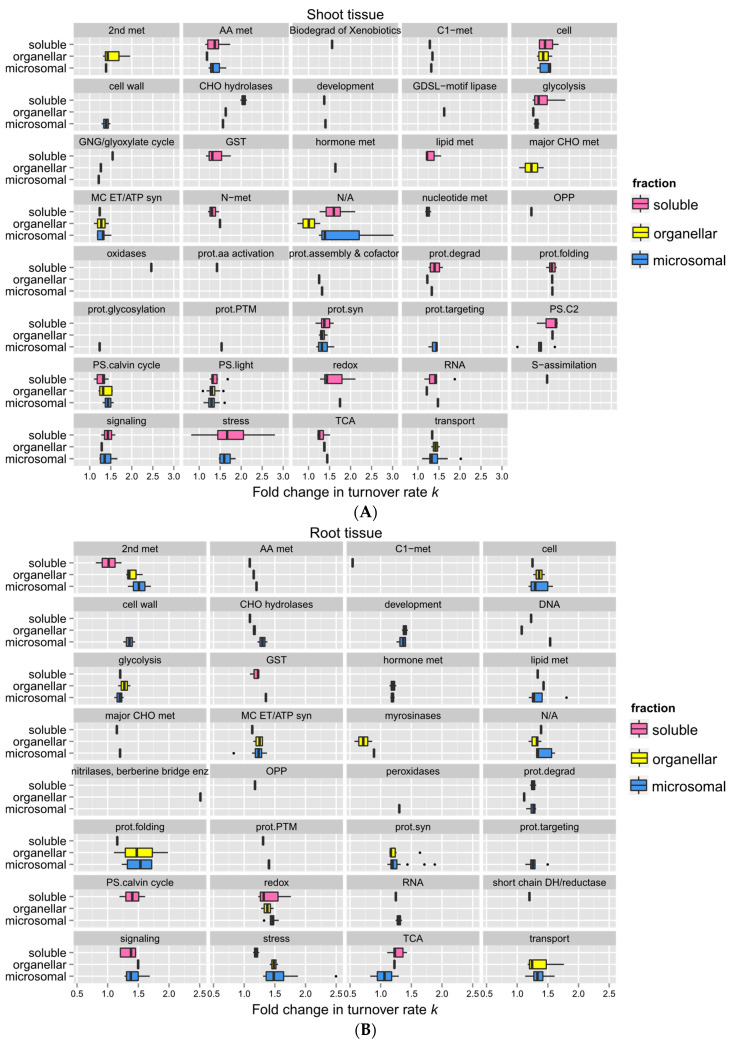
Comparison of protein functions with the change in turnover rates in response to 30 °C between different protein fractions**.** Boxes show the interquartile range (IQR) of the estimated fold change in protein turnover rate constant (*k*). Proteins are sorted in functional categorization, comparing results between the enriched soluble, organelle, and microsomal fractions of root (**A**) or shoot (**B**) tissues. The error bar represents the entire range of rates, and the closed circles represent outliers (1.5 IQR). N/A, protein function not assigned.

**Table 1 ijms-25-05882-t001:** The 10 fastest and lowest turnover proteins in the enriched soluble or membrane fraction of *Arabidopsis* roots ^a^.

	ID ^b^	Protein	AGI ^c^	Fraction ^d^	Turnover Rate ^e^	SD ^f^	Functional Category ^g^
Fastest	Q9M0A7	Putative uncharacterized protein (Gamma-glutamyl peptidase 1)	At4g30530	S	−4.397	0.0238	nucleotide met
	A8MRQ4_A8MSB9_F4JTU2_Q9SVM8	Glycine-rich RNA-binding protein 2, mitochondrial	At4g13850	S	−4.539	0.1128	RNA
	P20649	ATPase 1, plasma membrane-type	At2g18960	M	−4.607	0.0528	transport
	Q9SYM5	Trifunctional UDP-glucose 4,6-dehydratase/UDP-4-keto-6-deoxy-D-glucose 3,5-epimerase/UDP-4-keto-L-rhamnose-reductase RHM1	At1g78570	M	−4.624	0.0156	cell wall
	F4KIM7_Q9C5N2	Endomembrane family protein 70	At5g25100	M	−4.651	0.0223	N/A
	F4J1V2_Q94AW8	Chaperone protein dnaJ 3	At3g44110	M	−4.652	0.0881	stress
	P22953	Probable mediator of RNA polymerase II transcription subunit 37e (Heat Shock cognate Protein 70-1)	At5g02500	S	−4.668	0.0276	stress
	Q9XIE2	ABC transporter G family member 36 (AtABCG36)(PEN3)(PDR8)	At1g	M	−4.718	0.2347	transport
	P31414	Pyrophosphate-energized vacuolar membrane proton pump 1	At1g15690	M	−4.742	0.2033	transport
	Q9S791	Putative uncharacterized protein	At1g70770	O	−4.752	0.1353	N/A
Slowest	Q43348	Acid *beta*-fructofuranosidase 3, vacuolar (Vacuolar invertase 3)	At1g62660	S	−6.129	0.3853	major CHO met
	Q9C8Y9	*Beta*-glucosidase 22	At1g66280	O	−6.150	0.2910	CHO hydrolases
	P43297	Cysteine proteinase RD21a	At1g47128	M	−6.170	0.2050	prot.degrad
	P25819	Catalase-2	At4g35090	O	−6.176	0.4179	redox
	Q9FF53	Probable aquaporin PIP2-4 [Cleaved into: Probable aquaporin PIP2-4, N-terminally processed]	At5g60660	M	−6.227	0.0228	transport
	P46422	Glutathione S-transferase F2	At4g02520	S	−6.244	0.0349	GST
	A8MR01_F4JR94_O_2_3179	Patatin-like protein 1 (AtPLP1)	At4g37070	M	−6.245	0.5113	development
	Q9LHB9	Peroxidase 32	At3g32980	M	−6.261	0.2944	peroxidases
	Q9SIE7	Putative uncharacterized protein (PLAT-plant-stress domain-containing protein)	At2g22170	S	−6.314	0.0765	N/A
	Q9LTQ5	TRAF-like family protein	At3g20370	O	−6.320	0.3226	N/A
	Q9C8Y9	*Beta*-glucosidase 22	At1g66280	M	−6.594	0.5007	CHO hydrolases

^a^ Complete list in Appendix A. Only proteins with at least two unique peptides were used to calculate protein turnover rates. ^b^ Protein accession number assigned by the UniProt database. ^c^ The gene identification number assigned by the *Arabidopsis* genome initiative. ^d^ Enriched protein fractions: microsomal (M) fraction from the differential centrifugation (1 h, 100,000× *g*, pellet) of *Arabidopsis* root or shoot tissue homogenate; organelle (O) fraction from the differential centrifugation (5 min, 1500× *g*, pellet) of *Arabidopsis* root or shoot tissue homogenate; soluble (S) fraction from the differential centrifugation (1 h, 100,000× *g*, supernatant) of *Arabidopsis* root or shoot tissue homogenate. ^e^ The log_2_ value of protein turnover rate constant (*k*). ^f^ Standard deviation of protein turnover rate (log_2_*k*). ^g^ The functional category was adapted from the MapCave website [40].

**Table 2 ijms-25-05882-t002:** The 10 fastest and slowest turnover proteins in the enriched soluble or membrane fraction of *Arabidopsis* shoots ^a^.

	ID ^b^	Protein	AGI ^c^	Fraction ^d^	Turnover Rate ^e^	SD ^f^	Functional Category ^g^
Fastest	B9DG18_Q42547	Catalase-3	At1g20620	S	−4.479	0.1605	redox
	Q9CA67	Geranylgeranyl diphosphate reductase, chloroplastic	At1g74470	M	−4.746	0.1219	2nd met
	Q9CA67	Geranylgeranyl diphosphate reductase, chloroplastic	At1g74470	O	−4.857	0.1659	2nd met
	P56761	Photosystem II D2 protein	AtCg00270	M	−4.979	0.1141	PS.light
	P56761	Photosystem II D2 protein	AtCg00270	O	−4.986	0.0366	PS.light
	P56778	Photosystem II CP43 reaction center protein	AtCg00280	M	−5.101	0.1626	PS.light
	P56778	Photosystem II CP43 reaction center protein	AtCg00280	O	−5.127	0.0665	PS.light
	P42761	Glutathione S-transferase F10 (GST class-phi member 10)	At2g30870	S	−5.168	0.3743	GST
	Q9LKR3	Mediator of RNA polymerase II transcription subunit 37a (Heat Shock Protein 70-11)	At5g28540	M	−5.201	0.4357	stress
	P27202	Photosystem II 10 kDa polypeptide, chloroplastic	At1g79040	M	−5.220	0.2322	PS.light
	Q9LJG3	GDSL esterase/lipase ESM1	At3g14210	O	−5.261	0.0880	2nd met
	O80860	ATP-dependent zinc metalloprotease FTSH 2, chloroplastic	At2g30950	O	−5.307	0.1091	prot.degrad
	O80860	ATP-dependent zinc metalloprotease FTSH 2, chloroplastic	At2g30950	M	−5.312	0.1564	prot.degrad
	Q9SRV5	5-methyltetrahydropteroyltriglutamate--homocysteine methyltransferase 2 (AtMS2)	At3g03780	S	−5.356	0.2480	AA met
Slowest	O80934	Uncharacterized protein, chloroplastic	At2g37660	S	−6.783	0.2293	N/A
	Q8LE52	Glutathione S-transferase DHAR3, chloroplastic	At5g16710	S	−6.816	0.1549	redox
	P25857	Glyceraldehyde-3-phosphate dehydrogenase GAPB, chloroplastic	At1g42970	M	−6.861	0.1967	PS.calvin cycle
	Q9XFT3-2	Oxygen-evolving enhancer protein 3-1, chloroplastic (OEE3)	At4g21280	M	−6.928	0.2714	PS.light
	Q9SR37	*Beta*-glucosidase 23	At3g09260	O	−7.200	0.2308	CHO hydrolases
	Q9SR37	*Beta*-glucosidase 23	At3g09260	M	−7.218	0.2027	CHO hydrolases
	Q8W4H8	Inactive GDSL esterase/lipase-like protein 23 (Probable myrosinase-associated protein GLL23)	At1g54010	O	−7.438	0.1398	2nd met
	Q9SR37	*Beta*-glucosidase 23	At3g09260	S	−7.684	0.6082	CHO hydrolases
	Q9LXC9	Soluble inorganic pyrophosphatase 6, chloroplastic (PPase 6)	At5g09650	S	−7.774	1.5988	nucleotide met
	Q9LTQ5	TRAF-like family protein	At3g20370	O	−7.976	0.2116	N/A
	Q93Z83	TRAF-like family protein	At5g26280	O	−8.472	0.5887	N/A
	F4IB98	Jacalin-related lectin 11	At1g52100	O	−8.879	1.2147	hormone met

^a^ Complete list in Appendix A. Only proteins with at least two unique peptides were used to calculate protein turnover rates. ^b^ Protein accession number assigned by the UniProt database. ^c^ The gene identification number assigned by the *Arabidopsis* genome initiative. ^d^ Enriched protein fractions: microsomal (M) fraction from the differential centrifugation (1 h, 100,000× *g*, pellet) of Arabidopsis root or shoot tissue homogenate; organelle (O) fraction from the differential centrifugation (5 min, 1500× *g*, pellet) of Arabidopsis root or shoot tissue homogenate; soluble (S) fraction from the differential centrifugation (1 h, 100,000× *g*, supernatant) of Arabidopsis root or shoot tissue homogenate. ^e^ The log_2_ value of protein turnover rate constant (*k*). ^f^ Standard deviation of protein turnover rate (log_2_*k*). ^g^ The functional category was adapted from the MapCave website [40].

## Data Availability

The data presented in this study are available in Appendix A (doi: 10.5281/zenodo.11111496). The mass spectrometry proteomics data have been deposited to the ProteomeXchange Consortium (http://proteomecentral.proteomexchange.org, accessed on 23 May 2024) via the MassIVE partner repository with the data set identifiers PXD052532, PXD052528, PXD052530, PXD052533, PXD052525, and PXD052520.

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
