# Peer review of "Elevated Temperature Effects on Protein Turnover Dynamics in Arabidopsis thaliana Seedlings Revealed by 15N-Stable Isotope Labeling and ProteinTurnover Algorithm"

_ijms, 2024, doi:10.3390/ijms25115882_

Round 1

Reviewer 1 Report

Comments and Suggestions for Authors

In the manuscript named “Elevated Temperature Effects on Protein Turnover Dynamics in Arabidopsis thaliana Seedlings Revealed by 15N-Stable Isotope Labeling and Protein Turnover Algorithm”, Kai-Ting Fan and Yuan Xu have elevated temperature on proteome dynamics in Arabidopsis thaliana seedlings using 15N-stable isotope labeling and ultra-performance liquid chromatography-high resolution mass spectrometry. They have found that significant turnover alterations occurred with redox signaling, stress response, protein folding, et al, and which would helpful for plant heat tolerance. The findings were interesting to reads, and their conclusions were useful for molecular research about plant response to abiotic stress, the manuscript was also well prepared.

 (1) The manuscript has some data deposited online, please add links in manuscript.

(2) The section 2, line 117 to line 124 is suitable here?

(3) The figure 1 is no high-resolution in plot, please use high quality plot.

(4) In section 2.2.1, please support each p-value for comparison, as line 220 described. In addition, from line 181 to line 188, this discussion would be more suitable in discussion section.

(5) In section 2.3.1, many protein functions couldn’t be directly linked to figure 6 in authors description, please use functional categorization when they describe the proteins. In addition, the N/A should be explained in figure note. There were many categorizations with only one protein, which would have effects on their conclusion. The figure 6A and 6B could adopt some categorizations and some orders, which would be easily for comparing results.

(6) similar comments were about figure7 and figure 8, the terms with one protein, the results would be hardly for accepted, please revise these findings or description.

Reviewer 2 Report

Comments and Suggestions for Authors

The authors investigated the effects of elevated temperature on proteome dynamics in Arabidopsis thaliana seedlings using 15N-stable isotope labeling and ultra-performance liquid chromatography-high resolution mass spectrometry, coupled with the Protein Turnover algorithm. Results reported significant changes in the turnover rates of proteins, suggesting a balance between proteome stability and adaptability. 

General concept comments
The manuscript is relevant for the field, but it needs some changes. The introduction is too long. The experimental design is appropriate, and the results are reproducible. The discussion should be improved. Data Availability Statement is correctly reported. References are not so recent and too much self-references by Yuan Xu are detected. Conclusions section is consistent with results.

Specific comments 

The introduction is too long, some studies on 15N-labeling can be moved to discussion section.

Lines 117-119 should be moved to introduction section and the authors can deeply highlight the novelty of the study. 

Figure 1 has low quality resolution.

The discussion should be improved and expanded, emphasizing the results and discussing all the points presented in the work.

“beta” should be italicized in whole text.

Comments on the Quality of English Language

Minor editing of English language required
